# Motivating Personal Climate Action through a Safety and Health Risk Management Framework

**DOI:** 10.3390/ijerph20010007

**Published:** 2022-12-20

**Authors:** Charmaine Mullins-Jaime, Jan K. Wachter

**Affiliations:** 1Department of Built Environment, Bailey College of Engineering & Technology, Indiana State University, Terre Haute, IN 47809, USA; 2Safety Sciences Department, John J. and Char Kopchick College of Natural Sciences and Mathematics, Indiana University of Pennsylvania, Indiana, PA 15705, USA

**Keywords:** climate change communication, climate action motivation, climate change perception, climate change risk management

## Abstract

Background: There is overwhelming evidence the impacts of climate change present a probable threat to personal health and safety. However, traditional risk management approaches have not been applied to ameliorate the crises. The purpose of this study was to assess the impact on personal motivation for action of a communication intervention that framed climate change as a safety issue that can be mitigated through a safety and health risk management framework. Participants’ perception of climate change in terms of its anthropogenicity, context and importance, perception as a personal threat, belief in the efficacy of human action, motivating drivers for action, knowledge of climate change impacts, perceived personal barriers to climate action, and short- and long-term preferences for mitigating actions were evaluated. In addition, this study assessed the role of personal worldview on motivation for climate action. Methods: Through an online survey instrument embedded with a communication/education intervention, data were collected from N = 273 participants. Pre and post-intervention responses were assessed using Wilcoxon signed-rank tests and descriptive statistics. A path analysis assessed the influence of anthropogenicity, personal impact, and human efficacy beliefs on participant motivation for action. Multi-regression analyses and descriptive statics were used to evaluate the role of worldview on participant motivation for climate action. Results: Personal motivation for action significantly increased post-intervention. Anthropogenicity, personal impact, and human efficacy beliefs were predictive of personal motivation. Those who prioritized climate change as a safety issue and those driven by a desire to protect current and future generations had higher levels of personal motivation, post-intervention. Knowledge of climate change increased, psychosocial factors as barriers to climate action decreased, and preferences for personal mitigating actions shifted towards more impactful choices post-intervention. Holding Egalitarian worldviews significantly predicted climate action motivation. Conclusion: Presenting climate change and climate action strategies via a traditional health and safety risk management context was effective in increasing personal motivation for climate action. This study contributes to the literature on climate change communication and climate action motivation.

## 1. Introduction

Climate change is viewed by most scientists as an existential threat to life on earth [1,2,3], yet actions to mitigate have been minimal. Assessments from the Intergovernmental Panel on Climate Change (IPCC) and the United States Fourth National Climate Assessment make very clear the projected impacts present a probable threat to the personal health and safety of individuals in the United States and across the globe [2,4]. This makes climate change a hazard to human health and safety. However, traditional health and safety risk management approaches, such as the use of a hierarchy of controls [5] and plan-do-check-act (PDCA) [6] continuous improvement processes, have not been applied to ameliorate the crises.

Worry about climate change has been linked to higher likelihood of support and adoption of climate action [7]. Framing concern for climate change around personal health has been suggested in the literature as a possible solution to motivate individuals to act [8,9], particularly among those with cautious or dismissive attitudes toward climate change [10]. Framing climate change and impacts in context of gain or loss outcomes and as proximal and distal concerns have been examined in the literature [11,12] where lower psychological distance corresponds with higher level of concern, providing information on more severe distal outcomes can raise individual level of concern, and framing climate change mitigation as a positive gain can improve attitudes toward mitigation. Communicating the scientific consensus on climate change has been found to be effective as a “gateway” to shifting beliefs about climate change and bolstering support for climate action [13]. However, there are limitations of informational approaches [14]. Some of the shortcomings of climate change communication have been that learning about climate change can feel like one has already done something and communication with too much emphasis on fear of the problem without offering solutions can stagnate action [15,16].

Perceptions of climate change have been studied in the literature [17,18,19] and there have been various national and international perception surveys conducted over the last several years. Beyond pragmatic barriers to climate change mitigation, psychological barriers, best described by Gifford’s “Dragons of Inaction” [20], exist among the general population. Perception surveys, such as Yale Program on Climate Change Communication and George Mason University Center for Climate Change Communication on the psychological state of Americans towards climate change, assess how individuals conceptualize climate change, including viewing it as an environmental issue, a scientific issue, an agricultural issue, a severe weather issue, a health issue, an ethical, a political, and/or an economic issue [21]. However, it has not been assessed in the context of a safety issue, despite it being a very real and probable threat to personal safety.

In the United States, only about half of Americans believe climate change is a personal threat to them [22]. This may partially explain personal inaction around the crises as belief in the presence of a hazard and its risk level (associated with perceived likelihood of occurrence of adverse consequences and the severity of these consequences) are precursors to responding to a hazard from a safety and health risk management context. Using this context, some of the worst effects of climate change can be prevented through a continuous improvement process using sound risk analysis and then applying a hierarchy of controls [5] commonly adopted in the field of safety and health management. However, precursor beliefs must be established first to enable acceptance and motivation for climate risk mitigation.

As core tenets of motivation, according to classic Expectancy Value Theory, individuals must value the outcome and believe in the efficacy of their actions to achieve the outcome [23]. Beliefs and values such as skepticism of anthropogenic climate change [24], cause and effect of climate change-related impacts [17], worry about climate change [7], beliefs around proximal and distal outcomes [11,12], and the efficacy of actions [8] have been noted in climate change perception research as important variables in shaping support for action. The authors contend personal motivation for climate action can be increased by establishing those precursor beliefs through education and communication on climate change using a safety and health risk management framework that emphasizes a continuous improvement process and adoption of a hierarchy of controls in selecting mitigating actions to not only reinforce value perceptions but to also strengthen perceptions of self-efficacy.

To respond to the climate crises like a safety and health hazard, one must first view it as a hazard that poses some level of perceived risk. This involves holding some fundamental values and beliefs including a personal motivation to protect themselves and/or those with whom they have concern. In addition, as tested in this study, they should 1. believe climate change is anthropogenic—a human-caused problem requiring human action as literature indicates skepticism over anthropogenic climate change as a barrier to climate science acceptance and action [13,24,25]; 2. perceive personal risk—believe climate change affects them (recognition of the probability and severity of adverse effects) as literature indicates viewing the impacts as proximal issues and/or a health concern are important influences on motivation for action [7,9,10,11,12,17,25]; and 3. believe in the efficacy of human actions to address the climate crises as expectancy and instrumentality aspects of motivation [23,25].

Single token behavior is a common barrier to meaningful change [20] where people may take one positive action but stop there thinking it is enough or, worse, end up emitting more carbon from another aspect of their life. Carbon neutrality is a journey more likely to be reached by a continuous improvement process where each positive step builds momentum upon the other. Continuous improvement based on a Plan-Do-Check-Act (PDCA) cycle is a universally adopted and effective method used in systems safety and in quality and organizational systems management. While there are many differences between operation of organizations and personal and family units, it is reasonable to expect applying this risk management method to one’s personal choices will see similar benefits. It takes several small recurring steps to build good habits. By seeking to continually improve, and opting for the most effective options where practicable first, via a hierarchy of controls, one can build upon small actions and set goals for more overall impactful mitigation.

Because it will take the collective action of people from all worldviews to meet the climate crises, communication about climate change must speak to all worldviews. Cultural worldviews, as described by Douglas and Wildavsky’s Cultural Theory of Risk [26] and Kahan’s Cultural Cognition Theory [27], are considered in this study. Both cultural theory of risk and cultural cognition theory assign cultural views into four quadrants: Hierarchical, Egalitarian, Individualist, and Communitarian. While there are hundreds of thousands or even millions of nuanced worldviews [28], the framework for this model was broken into four group for the purposes of a parsimonious model [28] and because the theory assumes that there are only four stable organizational forms and the four quadrants represent the polarity of worldviews or “the borders” [26] with the remaining as possible mixtures within the quadrants [28]. Cultural Cognition Theory attempts to measure cultural worldview along two scales: the Hierarchist/Egalitarian “grid scale” and the Individualist/Communitarian “group scale” [27]. The grid scale deals with role differentiation and distribution of goods, offices, duties, and entitlements. The group scale deals with social organization and how individuals gravitate to community or self-reliance. In this framework, egalitarians and communitarians can justify mitigation through regulation as they are more likely to worry about environmental risks whereas individualists are more likely to reject claims of environmental risk if it contradicts their value of free markets. Hierarchists are more concerned with maintaining traditional social norms [29].

The purpose of this study was to assess the influence of using a communication intervention based on a safety and health (risk) management framework to improve personal motivation for climate action. The communication intervention presented basic climate science facts, presented the outcomes of climate change through a health and safety context, and discussed adopting a health and safety risk management approach that encouraged selection of mitigating actions using a hierarchy from most effective to least effective and adoption of a continuous improvement process like those used in safety management systems where one would continually evaluate activities, assess their impacts, and make further improvements. In addition to measuring personal motivation for climate action pre and post-intervention, this study examined the impact of a climate change communication/education intervention on personal perceptions and beliefs of climate change that serve as precursors of risk perception and risk treatment. These include anthropogenicity belief, conceptualization of climate change as a health and safety issue, belief that climate change impacts an individual personally, and belief in the efficacy of human actions to resolve the crises. It was hypothesized these beliefs would predict personal motivation for climate action. It was also hypothesized the scores of participants’ motivation to act, anthropogenicity belief, belief that climate change impacts them, and belief in the efficacy of human actions, would be higher post-intervention.

Further, barriers to climate action, general knowledge of climate change and its impacts, and the type of preferred mitigating actions were assessed pre and post-intervention. Participants’ interest in learning more and becoming involved in climate action was also assessed. Finally, because “worldview” has been found to be the strongest predictor of climate science acceptance [29,30,31,32], participants’ worldview was assessed along two scales, the Hierachist/Egalitarian grid scale and the Individualist/Communitarian group scale, in accordance with the work of Douglas and Wildavsky [26] and Kahan [27] to understand the influence of worldview on climate action motivation and what, if any effect, the intervention might have had on shifting motivation for action among various worldviews.

## 2. Materials and Methods

### 2.1. Population and Sample

Data were collected from five hundred and forty-four participants using convenience sampling. Participants were recruited via email request from Indiana State University and Indiana University of Pennsylvania mailing lists and from the principal investigator’s social media posts via LinkedIn and Facebook. The data were screened to remove participant responses from anyone who spent less than 20 min on the survey. This study was approved by the Indiana University of Pennsylvania Institutional Review Board. Participants were not financially compensated for participation. Informed consent was obtained at the start of the survey. A priori sample size test, using G*Power 3.1.9.6, indicated a minimum sample size of 107 for multiple linear regressions with 0.05 alphas, a medium effect size of 0.15, and power of 0.95.

### 2.2. Data Collection

An original online survey instrument was used consisting of thirty-four questions including three demographic questions on participants’ country, state/province, and employment industry and eight worldview questions, see Appendix A. Participants were asked questions to characterize their worldview based on Cultural Cognition Theory [27] as well as questions to assess their belief that climate change is human-caused (anthropogenic), the extent of individuals’ knowledge of climate change impacts, belief that climate change personally affects them, the importance of the context in which they view climate change (as an environmental issue, a health and safety issue, a scientific issue, an agricultural issue, a political issue, an ethical issue, and an economic issue), whether they believe humans can resolve the climate crises, their motivation to take action, what major context motivates them to act (to protect health and safety of current and future generations, to protect ecosystems and wildlife, and to protect the economy), what are their perceived barriers to action, and what actions they are currently taking and are planning to take to prevent climate change.

After answering the pre-intervention questions, participants watched a 20 min intervention video. The pre-intervention questions on anthropogenicity belief, knowledge of climate change impacts, belief that climate change personally affects them, the importance of the context in which they view climate change, belief that humans can resolve the climate crises, their motivation to act, what major context motivates them to act, perceived barriers to action were then repeated. In addition, participants were asked what mitigating actions they were willing to take within the near term (over the next 90 days), and what actions they were willing to take at some time in the future. Participants were also asked if and how the intervention had changed their view and motivation for climate action, whether the information presented had improved their knowledge and awareness of climate change and climate change prevention, as well as their interest in learning more and becoming involved in climate action through their workplace and community.

A national perception survey “Climate Change in the American Mind” [21] conducted by the Yale Program on Climate Communication and George Mason University Center for Climate Change Communication included a survey question on constructs of how Americans “conceptualize” global warming (as an environmental, scientific, severe weather, agricultural, political, economic, health, or ethical/moral/humanitarian issue). The constructs for understanding and prioritizing climate change in the present study were based on this survey question. However, in this study, the construct “safety” was added, framed as a “health and safety issue”, and participants were asked to rank order these issues from most to least important to them.

To assess what major context motivates them to act, participants were provided a selection of three reasons why they might be motivated to mitigate climate change: “to protect the health and safety of current and future generations”, “to protect ecosystems and wildlife, and “to protect the economy”. They were asked to rank order these motivating drivers from most to least important to them.

Barriers to climate action were based on the literature, particularly Gifford’s [20] “Dragons of Inaction”, where questions were phrased to reflect common pragmatic and psycho-social barriers to taking climate action as follows: “I’m not sure what I can do that will make a difference”, “I can’t afford the carbon-free or carbon-neutral alternative right now”, “Alternatives do not exist or are not practical for my lifestyle”, “I don’t think my family/social group/community group would approve”, “I have a conflict of interest in taking climate change action due to my employment, social or political affiliation”. Participants were asked to rank order these barriers from most to least important to them.

To assess knowledge of climate change impacts, participants were given thirteen items, based on current and projected impacts from IPCC reports [2] and the United States Fourth National Climate Assessment [4], and asked to select, both pre and post-intervention, all they believe are impacts of climate change. Changes in anthropogenicity belief, belief that climate change affects participants, efficacy belief, and personal motivation to act were measured by asking participants, both pre and post-intervention, to indicate their level of agreement from 0 to 100 where 0 is “not at all” and 100 is “strongly agree” with the following statements: “Climate change is primarily human-caused”, “Climate change affects me”, “Humans can resolve the climate crises”, and the motivation assessment question “I am ready to act now to mitigate (prevent) climate change”. To assess participant worldview, eight out of the 30 survey questions from Kahan’s Cultural Cognition as a Conception of the Cultural Theory of Risk [27] were used, four measuring the Hierarchist/Egalitarian scale and four measuring the Individualist/Communitarian scale.

The survey was pilot-tested for face validity among a small sample of participants, N = 8, to ensure the survey questions were understood as intended. The pilot-testing confirmed the survey questions and intervention video were understood. Minor changes were made to survey questions to ensure clear language.

### 2.3. Intervention

The intervention was embedded in the electronic survey and shown to participants after they completed the preliminary questions. The communication/education intervention was comprised of a 20 min video, created and presented by the principal investigator and showcased basic climate science facts and how climate change poses a health and safety threat to individuals in North America and around the world. A pathway to mitigation was presented through a safety management method in which a continuous improvement process using a hierarchy of controls can be used for a personal mitigation strategy.

The intervention attempted to galvanize action motivators found in the literature while addressing action barriers described by Gifford’s “Dragons of Inaction” [20] including “limited cognition”, “ideologies”, “social comparisons”, “sunk costs”, “discredence”, “perceived risks”, and “limited behavior”. The intervention was broken into two parts:

#### 2.3.1. Frame the Problem for Relevance and Immediacy to Individuals from Various Worldviews through the Lens of a Safety Issue

The presentation video provided clear and concise information about climate change including the 97% scientific consensus on climate change [1,13] and basic climate science facts through graphs and pie charts as the literature indicates presenting these facts have been effective in improving climate science acceptance [29,31,32,33]. Information was presented on how climate science has been presented in media and other public forums as if it were a political debate, some intentional by special interests [31,32,34,35,36,37,38], some simply as a media practice of giving equal attention to opposing sides [39,40], which can lead the general population to believe climate change is not a real concern or that its impacts are not as harmful as the scientific community projects. In addition, information was presented on how climate change poses a threat to personal health and safety, and a pathway to mitigation was presented through a safety management framework in which a continuous improvement process using a hierarchy of controls (Figure 1 and Figure 2) can be used for mitigation strategies as discussed below.

#### 2.3.2. Provide Clear Guidance on How Individuals Can Effectively Act to Mitigate Climate Change That Appeals to Various Worldviews through a Risk Management Approach Using an Adapted Hierarchy of Controls and Continuous Improvement Process

The intervention showed an example of a risk assessment tool based on likelihood of occurrence and severity of consequence, and the appropriate risk treatment guidelines based on the risk level, as well as the hierarchy of controls in selecting the mitigating actions as is commonly used in occupational health and safety management. It was then discussed that if the effects of climate change (fatality, heat and other illness, property damage, etc.) were happening in industry, in any first world country, the organization would conduct a risk assessment, carefully assess their activities that are contributing to the risk, assess the likelihood and severity of those risks and, if the risks were high or unacceptable, they would implement mitigating actions right away. There would not be much hesitation or debate as the risk assessment has been established and the only appropriate option is to mitigate, usually through control selection via a hierarchy of controls. It was then presented that this same approach should be taken for climate change mitigation.

Drawing on a traditional NIOSH hierarchy of controls method [5] used in health and safety management, the hierarchy presented in the intervention was tailored to address climate change mitigation (see Figure 1) where greenhouse gas emission elimination should be considered first prior to selection of emission reduction or offset activities. A variety of solutions including both government and free market-based ones were presented as part of this hierarchical approach to control selection. The literature indicates presenting free-market solutions are a viable option for reaching climate change consensus [29,33,35].

The following four steps were presented: assess, act, communicate, repeat, as depicted in Figure 2, where a continuous improvement risk management approach was discussed. The first step encouraged participants to view climate change as a safety issue requiring personal action, to believe their actions are important, no matter how small, and to assess their current activities and how they impact greenhouse gas emissions. Because this is a continuous improvement process, every action is valuable so long as impacts are continually reevaluated and further incremental actions are taken toward achieving net-zero emissions. The second step is to act by applying the hierarchy of controls approach to all of their activities where they opt for carbon elimination choices first (such as emissions-free power and transport, if practicable) before considering carbon reduction and then sequestration activities.

The third step promoted communication. Sharing ideas, concerns, and climate citizenship practices with their family, peers, employer, and elected representatives were presented as important actions individuals can take to on climate change. This step is critical as people are more inclined to accept information about risk and danger when it comes from someone who shares their values than when it comes from someone who holds opposing commitments [29]. However, according to the latest report on Climate Change in the American Mind, about six in ten Americans “rarely” or “never” discuss climate change with family and friends [22]. Given the effectiveness of peer communication in climate science acceptance and espoused values being a foundational element of culture [41], promoting espoused values on climate action is necessary to shift culture toward acceptance and action. Finally, the last step is to repeat this process, continually assessing and addressing carbon-emitting and carbon-demanding activities until one’s footprint is zero.

### 2.4. Data Analysis

Kendall’s Coefficient of Concordance and descriptive statistics were used to evaluate participants’ prioritization of constructs for understanding climate change, motivating drivers to mitigate climate change, and barriers to action. Rank order selections of constructs for understanding climate change and motivating drivers were then compared with mean motivation scores both pre and post-intervention.

Changes in anthropogenicity beliefs, belief that climate change affects individuals personally, efficacy beliefs, and personal motivation to act were measured using Wilcoxon signed-rank tests. A path analysis was used to assess if anthropogenicity belief, belief that climate change personally affects participants, and efficacy beliefs predict personal motivation to act, pre and post-intervention.

Changes in knowledge and awareness of climate change, selection of preferred mitigating actions pre and post-intervention, and participant interest in learning more and becoming involved in climate action through their community and workplace, were assessed descriptively. Multi-regression analysis was used to assess pre and post-intervention worldview orientation as a predictor of motivation to mitigate climate change.

Changes in motivation scores associated with participants’ worldviews were also assessed descriptively for any effect the intervention might have had on personal motivation to act. IBM SPSS versions 27 and 29, and AMOS 28 were used to analyze the data.

## 3. Results

Of the 544 participants, 273 spent longer than 20 min on the survey. Of the 273 participants, N = 267 fully answered the pre-intervention questions and N = 241 fully answered both pre and post-intervention questions, and are the basis of this analysis. 

### 3.1. Demographics

Table 1 shows participant country, region, and industry in which they work.

### 3.2. Motivation and Precursors of Motivation

#### 3.2.1. Importance of Frameworks for Understanding and Prioritizing Climate Change

When participants were asked to rank order the frameworks used for perceiving the issue of climate change (scientific, environmental, ethical, health and safety, agricultural or economic issue) from most to least important to them, there was significant fair agreement among raters with Kendall’s W = 0.285 pre-intervention and 0.351 post-intervention. When comparing pre and post-intervention rank order frameworks with pre and post-intervention mean motivation to act scores, pre-intervention results show ranking climate change as an ethical issue (for those that ranked it as the most important) had the highest mean motivation score at 81.39, followed closely by ranking climate change as a safety and health issue as their top priority with mean motivation score of 81.07. Post-intervention, ranking health and safety as the most important framework had the highest mean motivation score at 84.76, followed closely by those that ranked ethics as the most important framework, with a mean motivation score of 83.40. The mean motivation score changed from 81.07 pre-intervention to 84.76 post-intervention when climate change was ranked first as a health and safety issue, indicating the framework used for understanding and prioritizing climate change does matter in pre and post-intervention motivation to act. The safety and health framework for prioritizing climate change appears to be the best motivating framework, post-intervention. Table 2 shows post-intervention comparison of rank-ordered frameworks with participant mean motivation scores.

#### 3.2.2. Importance of Major Motivating Drivers for Climate Action Motivation

When participants were asked to rank order, from most to least important, major motivating drivers (reasons) to mitigate climate change, there was significant moderate agreement among raters (Kendall’s W = 0.576 pre-intervention and = 0.563 post-intervention). The selection “to protect the health and safety of current and future generations”, a safety and health motivation, was the top choice among respondents with mean ranks of 1.43 both pre and post-intervention. Table 3 shows post-intervention comparison of rank-ordered motivating drivers with participant mean motivation scores.

#### 3.2.3. Anthropogenicity Belief

Belief that climate change is human-caused (anthropogenic) was assessed based on the rationale that in order to be motivated to mitigate climate change, one must believe that it is a human-caused problem requiring human action for solutions. A Wilcoxon signed-rank test showed median anthropogenicity belief scores were higher post-intervention. This was a statistically significant increase (Z = −7.097, *p* < 0.001) in participant belief that climate change is human-caused.

#### 3.2.4. Belief That Climate Change Affects Participants

Belief that climate change affecting participants is a precursor of motivation from a safety context was tested in this study. Belief that one is being impacted by climate change or that climate change is a personal threat is critical to treating the climate crisis as a safety hazard that must be mitigated. Participant belief that climate change personally affects them increased post-intervention. A Wilcoxon signed-rank test showed a significant increase (Z = −9.310, *p* < 0.001) in the participants’ belief that climate change affects them.

#### 3.2.5. Belief in the Efficacy of Human Action

Efficacy, believing that humans can resolve the crises, is another critical precursor of motivation tested in this study. A Wilcoxon signed-rank test showed median efficacy belief scores were higher post-intervention. This was a statistically significant increase (Z = −10.533, *p* < 0.001) in participant belief that humans can resolve the climate crises.

Overall, in terms of better understanding the effects of beliefs on motivation among the participant population, the mean anthropogenicity score was high both pre-intervention (72.15) and post-intervention (79.21). However, there was an increased anthropogenicity belief after the intervention. Fewer participants initially held the belief that climate change affected them or that humans can resolve the crises prior to the intervention with mean scores of 46.95 and 65.41, respectively. The intervention was effective in increasing belief that climate change affects participants and that humans can resolve the crises with post-intervention mean scores of 83.41 and 81.29, respectively.

#### 3.2.6. Motivation

When testing the change in personal motivation for climate action, mean motivation scores changed from 68.25 to 73.91. A Wilcoxon signed-rank test of pre and post-intervention motivation scores showed significantly higher post-intervention median motivation scores (Z = −5.913, *p* < 0.001), indicating the intervention was effective in increasing participants’ motivation to mitigate climate change.

Results of the post-intervention Likert-type question asking if the information presented in this study had a positive, neutral, or negative impact on participants’ motivation to mitigate climate change showed 52.3% of participants indicated the information presented positively changed their motivation (22.3% “very positively”, 30% “positively”), 34.5% indicated the information did not change their motivation (23.1% indicated no change because they were already highly motivated and 11.4% indicated no change for other reasons), and 1.5% indicated the information presented negatively changed their motivation to mitigate climate change (0.4% “negatively”, 1.1% “very negatively”).

Figure 3 and Figure 4 show precursor beliefs for climate action motivation (climate change is human-caused, it personally affects the participant, and belief that humans can resolve the crises) significantly predicted motivation to mitigate climate change both pre-intervention, F(3, 263) = 142.824, *p* < 0.001, R2 = 0.62 and post-intervention F(3, 237)= 135.375, *p* < 0.001, R2 = 0.63. The model explained 62% of the variance in pre-intervention motivation for climate action and 63% of the variance in post-intervention motivation for climate action. Anthropogenicity belief, belief that climate change personally affects participants, and the belief that humans can resolve the crises contributed significantly to the models with pre-intervention standardized coefficients as follows (β = 0.27, *p* < 0.001), (β = 0.41, *p* < 0.001), (β = 0.20, *p* < 0.001), and post-intervention standardized coefficients as follows (β = 0.24, *p* < 0.05), (β = 0.41, *p* < 0.001), (β = 0.19, *p* < 0.05).

### 3.3. Importance of Barriers to Climate Action

When participants were asked to rank order their most relevant barriers to climate action, there was significant moderate agreement among raters both pre- (W = 0.501) and post-intervention (W = 0.447). The top two barriers to climate action were pragmatic: “I can’t afford the carbon-free or carbon-neutral alternative right now” with a mean rank of 1.70 pre-intervention and 1.62 post-intervention, followed by “Alternatives do not exist or are not practical for my lifestyle” with a mean rank of 2.45 pre-intervention and 2.41 post-intervention. The third highest ranking barrier to climate action was knowledge: “I’m not sure what I can do that will make a difference” with a mean rank of 2.51 pre-intervention and 2.90 post-intervention. Fewer people selected this as a top barrier post-intervention indicating the intervention was effective in increasing their knowledge of what they can do to mitigate climate change.

The fourth-ranking barrier was social: “I don’t think my family/social group/community group would approve” with a pre-intervention rank of 4.09 and post-intervention mean rank of 3.84. The least important barrier to participants was conflict of interest with a pre-intervention mean rank of 4.26 and post-intervention mean rank of 4.22.

### 3.4. Selection of Preferred Mitigating Actions Pre and Post-Intervention

Table 4 shows a list of climate change mitigating actions and the percentage of participants that selected them as current actions, actions they would like to take over the next 90 days, pre-intervention and actions they would like to take over the next 90 days and actions they would like to take at some later time in the future, post-intervention. When comparing responses to the pre-intervention question on participants’ current actions with responses to the post-intervention question on what short-term actions they indicated they would take over the next 90 days, the actions more frequently selected, post-intervention, were: “Buy locally produced or low carbon alternative products and food”, “Use more low carbon forms of transportation”, “Grow/produce some of my own food”, “Buy less stuff that I don’t need”, “Speak with family, colleagues, and friends about reducing their carbon footprint”, and “Speak with my local elected representatives, (municipal, state/provincial, and federal) about my desire for climate change action”.

The long-term actions that saw the greatest increase post-intervention were “Install geothermal energy system on my home”, “Go 100% renewable for all my energy consumption needs”, “Install solar panels on my home”, “Speak with my employer to get them to take actions to reduce their carbon footprint”, and “Use more low carbon forms of transportation”.

### 3.5. Knowledge

Table 5 shows a comparison of the percentage of participants that indicated their awareness of various impacts of climate change pre and post-intervention. Awareness of all impacts increased post-intervention with the largest increases in awareness of the following impacts: increase in vector-borne illnesses and fatalities +24.4%, increase in violent conflict +20.5%, cause millions of people to become displaced +17.8%, cause trillions of dollars in property damage and loss +15.9%, and increase in heat-related illness and fatalities +15.0%.

Of 241 participants who responded to the post-intervention questions, 155 indicated the information presented improved their perceptions, knowledge, and awareness of climate change and its mitigation (62 “very much improved”, 93 “improved”), 84 participants indicated neutral responses (68 indicating they had previous knowledge and 16 indicating neutral response for other reasons) and 2 participants indicated the information presented decreased their perceptions, knowledge, and awareness of climate change and mitigation (1 “very much decreased”, 1 “decreased”).

### 3.6. Participant Interest in Learning More about How They Can Mitigate Climate Change

The majority of the 241 participants who answered the post-intervention questions indicated they were interested in learning more about climate change with 11.4% interested in learning more only through their workplace, 18.3% interested in learning more only through their community, and 51.6% of participants interested in learning more through both their community and workplace. This may suggest the general public is ready to receive messaging and is willing to engage through their community and workplace.

### 3.7. Worldview

Multiple regression analyses to predict participants’ pre and post-intervention motivation to mitigate climate change based on worldview grid and group scores were statistically significant pre-intervention F(2, 264) = 83.603, *p* < 0.001, adjusted R2 = 0.383 and post-intervention F(2, 238) = 58.635, *p* < 0.001, adjusted R2 = 0.324. In accordance with Cultural Cognition Theory [27], the grid scale ranged from −1, −0.5, 0, 0.5, 1 along the Y axis and was scored positively if responses indicated Hierarchist tendencies and negatively if responses indicated Egalitarian tendencies. Similarly, the group scale ranged from −1, −0.5, 0, 0.5, 1 along the x-axis and was scored positively if responses indicated Communitarian tendencies and negatively if responses indicated Individualist tendencies. Only the Hierarchist/Egalitarian scale (grid) statistically significantly predicted motivation to mitigate climate change both pre-intervention (β = −0.554, *p* < 0.001) and post-intervention (β = −0.521, *p* < 0.001). The group scale did not significantly predict climate action motivation pre-intervention (β = 0.091, *p* = 0.193) and post-intervention (β = 0.071, *p* = 0.354). Since the grid regression coefficients were negative and significant, those whose worldview fell into the Egalitarian attitudinal range on the grid had higher motivation to take climate action and those who fell into the Hierarchist range had lower motivation both pre and post-intervention.

When grouping participant worldview into categories based on their grid and group scores, many participants provided neutral responses in one or both scales making categorization into 4 quadrants impractical. Based on their responses to the Individualist/Communitarian and Hierarchist/Egalitarian scales, individuals were grouped into the following nine categories: Communitarian/Egalitarian, Communitarian/Hierarchist, Communitarian/Neutral, Individualist/Egalitarian, Individualist/Hierarchist, Individualist/Neutral, Neutral/Egalitarian, Neutral/Hierarchist, and Neutral/Neutral. The majority of participants (110 of the 267 respondents who answered worldview questions) were in the Neutral/Neutral category, 53 participants were in the Communitarian/Neutral category, 26 were categorized as Individualist/Neutral, 23 were categorized as Communitarian/Egalitarian, 18 were categorized as Neutral/Egalitarian, 12 were categorized as Individualist/Egalitarian, 10 as Individualist/Hierarchist, 10 as Neutral/Hierarchist, and 5 as Communitarian/Neutral, an approximately normal distribution. The mean motivation scores for all worldview categories increased post-intervention, see Table 6.

## 4. Discussion

The safety and health framework for prioritizing climate change appears to be the best motivating framework, post-intervention, followed by viewing it as a moral/ethical issue. Interestingly, the lowest action motivation scores were associated with those participants who prioritized climate change as a political issue which is consistent with the literature. When scientific issues are publicized in the media, the media often takes a “fairness doctrine” approach where these scientific issues are treated as political issues, as if there are equal opposing sides and each side of the debate is provided equal coverage [42]. This approach misrepresents the scientific consensus and creates a vacuum in which denial to satisfy cognitive dissonance [43] with personal worldview can thrive.

More participants prioritized climate change as an environmental issue both pre and post-intervention which is consistent with the notion of environmental stewardship as a widely adopted value in which people act because the environment is relevant to each of us [44]. However, environmental protection for the sake of stewardship allows for passivity that can be counterproductive to addressing the climate crises. This study sheds light on the mechanics of personal value systems where environmental ethics is a powerful motivator. However, the level of motivation for action is surpassed if climate change is prioritized as a personal health and safety issue.

The selection of safety and health as a primary driver for action was also associated with higher levels of personal motivation to mitigate climate change, compared to selection of environmental reason and economic reason, Table 3. This may reflect ensuring a healthy and safe legacy for future generations is a powerful lens through which the climate change crisis is being viewed and as a motivator for climate change action.

Anthropogenicity belief, belief that climate change affects participants, and belief that humans can resolve the climate crises were predictive of motivation to take mitigating action, both pre-and post-intervention. These precursor belief scores and participants’ motivation to mitigate climate change scores significantly increased post-intervention. The results are consistent with Vroom’s Expectancy Value Theory [23], where motivation is determined by how likely the behavior will result in the desired outcome (“expectancy”) and how much the individual “values” the desired outcome. Later expansions of Expectancy Value Theory explain values in context of Maslow’s Hierarchy of Needs [45] where “value” is determined by the extent the outcome satisfies the fundamental physiological needs, the needs for safety and security, the need for autonomy, and self-actualization [46]. The intervention was effective in shifting individual beliefs of climate change as a safety issue, one that personally affects them (safety context) and driven by a desire to protect people (safety context). The corresponding increase in motivation further supports the premise that climate change as a safety and health issue is a fundamental variable and a top-tier “value” if explaining motivation through Expectancy Value Theory. Interestingly, the greatest change in precursor belief scores can be understood from a personal safety context. “Climate change affects me” can be viewed as a perceived personal threat and further supports the crux of this study that climate change should be framed as a personal safety issue and mitigated through a safety/risk management context.

The results also showed increased belief in the efficacy of humans to resolve the climate crises and a corresponding increase in motivation for action. This supports “expectancy” in climate mitigating actions as an important variable in motivating individuals to act. The intervention provided a risk management framework based on well-established risk management principles, such as hazard mitigation via a hierarchy of controls and a continuous improvement process, which may have helped improve “expectancy”, since this method is well-established in industry and was presented as being effective in hazard and risk mitigation.

Climate change will threaten our ability to meet fundamental needs including basic physiological and personal safety and security needs that must be met and will take precedence over others. The trouble is many Americans do not perceive climate change as a safety threat or something that affects them personally [21,22]. The intervention in this study was found to be effective in shifting that belief. Framing climate change as a health and safety issue personalizes and gives value and immediacy to the crises. Moreover, presenting the use of a risk management framework to mitigate the crises, based on a continuous improvement model, can help create expectancy in the outcomes and appear to be important in stimulating motivation.

The increase in selecting pragmatic barriers post-intervention and decrease in psychosocial barriers may signal the intervention was effective in its intent to disentangle practical responses to climate change from personal worldview and social constructs as barriers. The intervention presented mitigation through a hierarchy of controls where carbon elimination choices are most effective, which are often pragmatic barriers due to cost. For example, emission-free alternatives, such as installing solar or geothermal systems on homes or opting for an electric vehicle, are cost-prohibitive or not practical for many. Pragmatic barriers require pragmatic solutions, an indication that additional support from government and industry is needed to address the economics and practicality of top-tier carbon mitigation options. However, government and industry support are entirely dependent on the wants and needs of its electorate and customers, respectively. Thus, educating the populace on the importance of climate change mitigation and encouraging more open discussion of concerns, as promoted in step three of the intervention hazard mitigation process, is a crucial step to creating a groundswell of support for climate change mitigation.

When comparing pre and post-intervention responses to question on participants’ preferred actions, many of the most impactful long-term actions that fell under the “carbon elimination” category (vs. reduction or sequestration categories) saw the greatest increase in selection post-intervention indicating this communication framework can be an effective tool in getting the general public to recognize the value of top-tier (e.g., “elimination”) climate mitigating actions.

By providing information on the types of mitigating actions that would be most effective and by presenting mitigation as a multi-step continuous improvement process, the intervention attempted to address the knowledge problems of limited cognition and limited behavior or “tokenism” [20]. Consequently, participants tended to select more effective types of mitigating actions, post-intervention, an indication the intervention may have been successful in addressing the “limited cognition” dragon of inaction [20] through increasing knowledge of effective mitigating actions and the “limited behavior” problem by promoting the continuous improvement process to prevent stagnation.

Fewer people selected lack of knowledge as an important barrier to action, post-intervention. While knowledge did improve after watching the intervention video, it remained in the top three barriers to action. The phrase used for assessing knowledge as a barrier was “I’m not sure what I can do that will make a difference” and supports the idea that knowledge of the efficacy of actions plays a significant role in the barriers to action and further steps are needed to shift beliefs in the efficacy of personal actions.

Knowledge has been identified as a critical barrier to climate change action [20]. The intervention attempted to raise awareness of both the impacts of climate change and the importance of taking mitigating action. It is possible the new information created cognitive dissonance [43] where the consequences of climate change were viewed as a threat to worldview. Thus, participants may have been more motivated to reduce this dissonance by indicating an increased level of motivation for climate action.

Concerning worldview grid and group scales, the grid scale deals with role differentiation and how goods, offices, duties, and entitlements are distributed. Hierarchists or those with a “high grid” way of life tend to support social status or classification based on age, race, gender, bureaucratic office held, lineage, etc., and will spend much time and attention protecting the rank and order that support their position and interests [27]. A “low grid” way of life or Egalitarian worldview tends to support the belief that no one is prevented from participation in any social role or status due to gender, age, race, or family connections [27].

Hierarchists might view climate action as a threat to their personal wealth or status, i.e., if that status is supported by burning fossil fuels, they may feel their position threatened by an abrupt reduction in fossil fuel consumption whereas the Egalitarian worldview is not dependent on these “rank-based” constraints [27]. Those with a strong Egalitarian worldview are more likely driven by environmental justice as they might perceive hierarchical distribution of entitlements as unfair and may be more motivated to address the social injustice associated with climate change.

While the Individualist worldview has been linked to dismissal of environmental concerns [26] including climate change [29], in this study, it seems climate change and climate action may be non-threatening to the core tenets of their values on both ends of the “group” scale as the Individualist/Communitarian scale was not a significant predictor of climate action motivation. A “weak group” way of life or an Individualist worldview supports competition and self-reliance. Those with “strong group” or a Communitarian worldview are more supportive of solidarism where people should be able to depend on the government and on one another and tend to believe that societal interests should take precedence over individual interests [29].

Information and topics that cause cognitive dissonance [43] with personal worldview may be greater change agents than information and topics that support worldview and might explain why the “grid” scale was the only predictor of motivation to mitigate climate change as the intervention presented many solutions that would have appealed to both individualist and communitarian worldviews. It is possible that if information presented in the intervention had clashed with the Individualist or Communitarian worldviews, such as proposing mitigation through carbon taxes or proposing the need for government interference, the “group” scale might have predicted motivation to act due to the threat to personal worldview.

Mean motivation scores among all worldviews increased, post-intervention. A possible explanation is the intervention discussed impact to society, economy, and human health and safety and presented a variety of climate change mitigating actions that could appeal to various worldviews. Promoting self-reliance through food and energy production would appeal to Individualists whereas collective strategies likely appealed to Communitarian worldviews. Hierarchists may have recognized that climate mitigation can act as a stabilizer in society, making mitigation more appealing, whereas information presented on disproportionate impacts to the disadvantaged likely appealed to those with Egalitarian views.

### Limitations

While this research contributes to the literature on climate change perceptions, there are limitations to be considered when interpreting the results of this study. Data were cross-sectional so causation cannot be inferred. Data were collected from a relatively small sample size (N = 241). There may be selection bias and, while the survey was anonymous, there may be social desirability bias in survey responses. Data were collected using convenience sampling techniques, recruiting participation via social media posts and the use of email lists, which may limit the generalizability of the study results. Political and financial interests of the respondents and their means to overcome action barriers are also limitations that might affect the generalizability of this study. However, when grouping worldview grid and group scales into categories, the distribution of worldview categories was approximately normal, which may support the generalizability of the results. The results likely reflect an American perspective vs. a global perspective, particularly a Midwestern perspective, as the majority of participants were from this region in the United States. Although participant responses were excluded from anyone who spent less than 20 min on the survey (a minimum time amount needed to fully experience the intervention), evaluating active viewing or engagement was not a capability of the online survey software. Replication of this study in a live setting or use of engagement tracking software in an online setting would control for this limitation.

## 5. Conclusions

This study intended to test the impact of a climate change communication intervention to promote participant motivation for climate change action by framing the climate crises as a health and safety problem that can be mitigated through a health and safety risk management method. The intervention was designed to appeal to a general audience and address the most common psychological barriers to action on climate change found in the literature. Participants who viewed climate change as a health and safety issue had the highest levels of motivation post-intervention. The intervention was effective in shifting perceptions of climate change as a safety issue, post-intervention where more participants prioritized it as a safety and health issue. The intervention was effective in increasing motivation to mitigate (prevent) climate change and shifted fundamental beliefs surrounding motivation that serve as precursors to risk recognition and treatment. These beliefs included anthropogenicity, perception of climate change as a personal threat, and belief in the efficacy of human actions.

This study showed framing climate change as a health and safety issue, one that personally affects individuals, is an effective motivator for climate action. The precursor beliefs of anthropogenicity, personal impact, and efficacy significantly predicted motivation for climate change mitigation. For something to be treated like a safety issue, it must first be viewed as a hazard with probable undesired outcomes determined and acknowledged in order to spur a mitigating response. The nearly inexhaustible scientific evidence on climate change support that consequences are a severe risk to personal health and safety. Thus, a safety management context is an appropriate framework for response. Perceiving climate change as a personal threat or “safety hazard” may be a necessary spark, and use of hierarchical control selection and continuous improvement methods may aid in sustaining the motivation needed to fuel the wildfire of dramatic changes needed to address the climate crises.

The communication intervention improved general knowledge and awareness of climate change and shifted the type of mitigating long-term actions participants indicated they would like to take towards more impactful choices of carbon elimination vs. reduction and sequestration. The intervention was effective in shifting beliefs and motivation, as well as increasing the motivation scores of individuals from all worldview categories. Thus, framing climate change as a health and safety problem that should be managed through health and safety management methods is an important framework that could be a linchpin in the sustainability movement.

Those wishing to increase personal climate action support should present the consequences as safety issues, ensure they raise awareness of the efficacy of mitigating actions, and present the fundamental value of hazard mitigation from a safety context, as was presented through the communication intervention in this study. It is recommended similar communication interventions be considered as part of public health and safety communication strategies on climate action. Future research should assess the effect of waning motivation and if particular elements of the communication intervention used in this study are more effective in shifting motivation, beliefs, and mitigation preferences over the short and long term.

## Figures and Tables

**Figure 1 ijerph-20-00007-f001:**
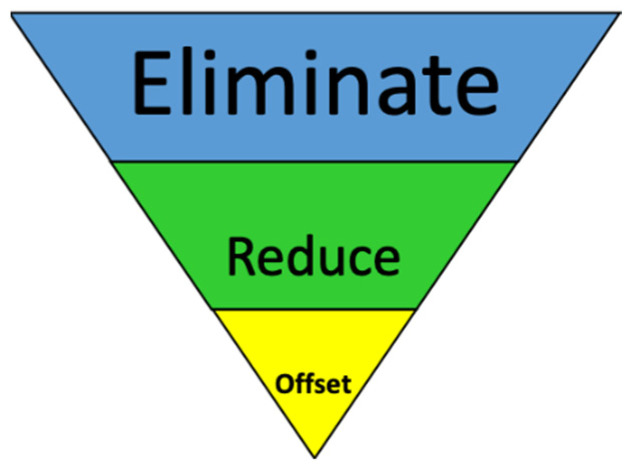
Adapted Hierarchy of Controls for Climate Change Mitigation.

**Figure 2 ijerph-20-00007-f002:**
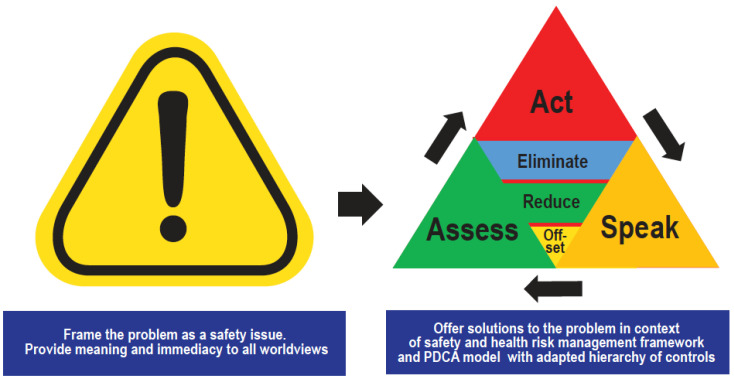
Climate Action Communication and Mitigation Strategy Through a Health and Safety Risk Management Framework.

**Figure 3 ijerph-20-00007-f003:**
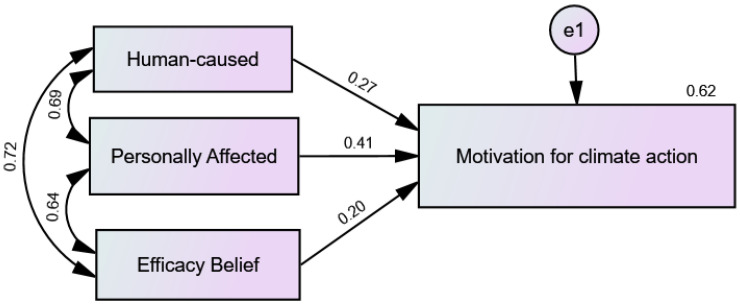
Impacts of Pre-intervention Climate Action Precursors on Pre-intervention Climate Action Motivation.

**Figure 4 ijerph-20-00007-f004:**
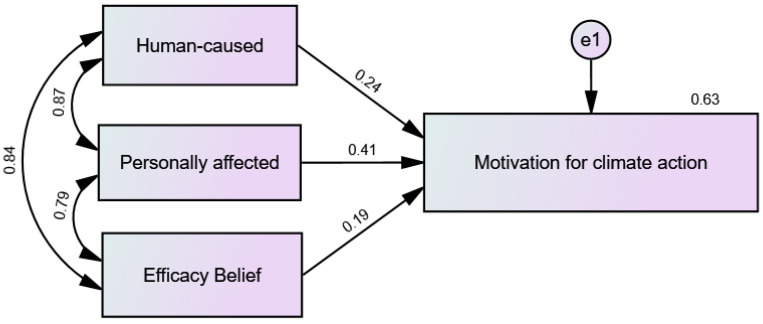
Impacts of Post-intervention Climate Action Precursors on Post-intervention Climate Action Motivation.

**Table 1 ijerph-20-00007-t001:** Demographics.

	N	%
*Country*		
United States of America	214	78.4%
Canada	19	7.0%
Other	5	1.8%
Missing	35	12.8%
Total	273	
*Region*		
Midwestern United States	142	52.0%
Northeastern United States	41	15.0%
Southeastern United States	19	7.0%
Western United States	6	2.2%
Southwestern United States	5	1.8%
Western Canada	7	2.6%
Atlantic Canada	6	2.2%
Eastern Canada	6	2.2%
Regions outside the United States and Canada	5	1.8%
Missing	36	13.2%
Total	273	
*Industry*		
Educational Services	65	23.8%
Construction	56	20.5%
Health Care and Social Assistance	18	6.6%
Government, Public Administration	13	4.8%
Manufacturing	12	4.4%
Retail	11	4.0%
Professional and Business Services	8	2.9%
Transportation and Warehousing	6	2.2%
Agriculture, Forestry, Fishing, Hunting	4	1.5%
Mining, Quarrying, and Oil and Gas extraction	3	1.1%
Utilities	2	0.7%
Other	41	15.0%
Missing	34	12.5%
Total	273	

**Table 2 ijerph-20-00007-t002:** Comparison of Post-intervention Motivation Scores with Post-intervention Top Ranking Frameworks for Conceptualizing Climate Change Issue.

Prioritization of Climate Change as an Issue	Post-Intervention Motivation Mean Score	N	Std. Deviation
Health and safety issue as top priority	84.76	83	23.88
Ethical/moral issue as top priority	83.40	15	24.83
Environmental issue as top priority	74.46	95	26.46
Economic issue as top priority	63.00	2	38.18
Scientific issue as top priority	56.96	25	35.48
Agricultural issue as top priority	51.12	8	33.45
Political issue as top priority	38.00	13	45.41

**Table 3 ijerph-20-00007-t003:** Comparison of Post-intervention Motivation Scores with Post-intervention Top Ranking Motivation Drivers.

Top Ranking Motivation Drivers	Post-Intervention Motivation Mean Scores	N	Std. Deviation
To protect the health and safety of current and future generations as top priority	77.99	145	27.51
To protect ecosystems and wildlife as top priority	70.55	88	32.10
To protect the economy as top priority	37.12	8	41.52

**Table 4 ijerph-20-00007-t004:** Comparison of Short-term and Long-term Mitigating Actions Pre- and Post-intervention.

	Pre-Int. Current Actions	Pre-Int. Planned Future Actions	Post-Int. Actions 0–90 Days	Post-Int. Actions at Some Time in the Future	Delta: Post-Int. Actions within the Next 0–90 days—Pre-Int. Current Actions	Delta: Post-Int. Future Actions—Pre-Int. Future Actions
Buy less stuff	70%	67%	79%	67%	9%	0%
Use less stuff/3rs	77%	67%	74%	66%	−3%	−1%
Buy local and low carbon products	47%	58%	63%	62%	16%	4%
Produce some of my own food	42%	55%	52%	60%	10%	5%
Change my light bulbs to LED	79%	64%	70%	66%	−9%	2%
Conserve energy	78%	69%	75%	66%	−3%	−3%
Use low carbon transportation	23%	43%	37%	50%	14%	7%
Buy/drive an EV	6%	42%	7%	44%	1%	2%
Install solar panels on my home	7%	44%	6%	52%	−1%	8%
Install geothermal energy system on my home	4%	23%	4%	36%	0%	13%
Go 100% renewable for all my energy consumption needs	5%	22%	7%	33%	2%	11%
Speak with my employer to get them to take actions to reduce their carbon footprint	14%	24%	20%	32%	6%	8%
Speak with family, colleagues and friends	30%	38%	39%	44%	9%	6%
Speak with elected representatives	9%	26%	18%	32%	9%	6%
None of these	3%	4%	4%	5%	1%	1%

**Table 5 ijerph-20-00007-t005:** Comparison of Pre and Post-intervention Knowledge of Climate Change Impacts.

Impacts of Climate Change	% Pre-Intervention Knowledge	% Post-Intervention Knowledge	Delta
More vector-borne illness and fatalities	57.3	81.7	24.4
Increase in violent conflict	55.4	75.9	20.5
Cause millions of people to become displaced	69.3	87.1	17.8
Cost trillions of dollars in property damage/loss	70.4	86.3	15.9
More heat-related illness and fatalities	70.4	85.4	15.0
Cause large parts of the Earth to become uninhabitable	70.0	84.2	14.2
Increase in severe floods	80.5	89.6	9.1
Have a negative effect on the economy	80.1	89.2	9.1
Food shortages and famine	79.4	87.1	7.7
Increase in severe droughts	84.6	90.9	6.3
Increase in wildfires	86.9	90.9	4.0
Have a negative effect on human health and safety	85.4	90.0	4.6
Increase in severe weather events	87.2	90.5	3.3

**Table 6 ijerph-20-00007-t006:** Comparison of Worldview Category with Pre and Post-intervention Mean Motivation.

Worldview Category	Pre-Mean	Post-Mean	Delta
Communitarian/Egalitarian	72.52	78.59	6.07
Communitarian/Hierarchist	71.80	86.80	15.00
Communitarian/Neutral	75.85	78.94	3.09
Individualist/Egalitarian	61.83	76.27	14.44
Individualist/Hierarchist	63.70	68.33	4.63
Individualist/Neutral	60.12	67.80	7.68
Neutral/Egalitarian	72.50	85.19	12.69
Neutral/Hierarchist	38.20	50.90	12.70
Neutral/Neutral	68.60	71.79	3.19
Total	68.25	73.91	5.66

## Data Availability

The data presented in this study are available on request from the corresponding author.

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
