# Peer review of "Motivating Personal Climate Action through a Safety and Health Risk Management Framework"

_ijerph, 2022, doi:10.3390/ijerph20010007_

Round 1

Reviewer 1 Report

This study aimed to assess the impact of a communication intervention on individual motivation for action that addresses climate change as a security issue that can be mitigated through a security and health risk management framework, along with an assessment of individual worldviews on motivation for climate action. The topic is interesting. Some comments for the authors to improve the quality of the manuscript are below.

1. In lines 34-35, please use the correct format of Keywords.

2. In lines 105-109, please add some references to support the sentences.

3. In line 118, how about the explanation of "Douglas and Wildavsky’s Cultural Theory of Risk "?

4. In line 174, please uniform the format of reference.

5. In lines 197-202, please try to simplify the sentence structure so as not to affect the reading experience.

6. In lines 291-300, it is better to write the three steps together or write them in sections.

7. In line 332, please indent before the paragraph.

8. In the Demographics section, it is better to introduce the participant's allocation criteria in the section of Participant.

9. Please uniform the format of Table 1.

10. Where is the limitation of the study? Please add in the section of Conclusions.

Author Response

Dear reviewer,

We would like to thank you for your valuable feedback. We believe your input has helped us to improve the manuscript. Below are responses to your feedback: 

  1. In lines 34-35, please use the correct format of Keywords.

Thank you for this suggestion. This has been updated.

  1. In lines 105-109, please add some references to support the sentences.

Thank you for this suggestion. The section has been updated as follows: In addition, as tested in this study, they should 1. believe climate change is anthropogenic— a human-caused problem requiring human action as literature indicates skepticism over anthropogenic climate change as a barrier to climate science acceptance and action [13,24,25] ; 2. perceive personal risk—believe climate change affects them (recognition of the probability and severity of adverse effects) as literature indicates viewing the impacts as proximal issues and/or a health concern are important influences on motivation for action [7,9-12,17,25]; and 3. believe in the efficacy of human actions to address the climate crises as expectancy and instrumentality aspects of motivation [23,25].

  1. In line 118, how about the explanation of "Douglas and Wildavsky’s Cultural Theory of Risk "?

Thank you for this suggestion. The section has been updated as follows: Because it will take the collective action of people from all worldviews to meet the climate crises, communication about climate change must speak to all worldviews. Cultural worldviews, as described by Douglas and Wildavsky’s Cultural Theory of Risk [26] and Kahan’s Cultural Cognition Theory [27], are considered in this study. Both cultural theory of risk and cultural cognition theory assign cultural views into four quadrants: Hierarchical, Egalitarian, Individualist, and Communitarian. While there are hundreds of thousands or even millions of nuanced worldviews [28], the framework for this model was broken into four group for the purposes of a parsimonious model [28] and because the theory assumes that there are only four stable organizational forms and the four quadrants represent the polarity of worldviews or “the borders” [26] with the remaining as possible mixtures within the quadrants [28]. Cultural Cognition Theory at-tempts to measure cultural worldview along two scales: the Hierarchist/Egalitarian “grid scale” and the Individualist/ Communitarian “group scale” [27]. The grid scale deals with role differentiation and distribution of goods, offices, duties, and entitlements. The group scale deals with social organization and how individuals gravitate to community or self-reliance. In this framework, egalitarians and communitarians can justify mitigation through regulation as they are more likely to worry about environmental risks whereas individualists are more likely to reject claims of environmental risk if it contradicts their value of free markets. Hierarchists are more concerned with maintaining traditional social norms [29].

  1. In line 174, please uniform the format of reference.

Thank you for this suggestion. This has been updated.

  1. In lines 197-202, please try to simplify the sentence structure so as not to affect the reading experience.

Thank you for this suggestion. The section has been updated as follows: A national perception survey "Climate Change in the American Mind" [21] conducted by the Yale Program on Climate Communication and George Mason University Center for Climate Change Communication included a survey question on constructs of how Americans “conceptualize” global warming (as an environmental, scientific, severe weather, agricultural, political, economic, health, or ethical/moral/humanitarian issue).  The constructs for understanding and prioritizing climate change in the present study were based on this survey question. However, in this study, the construct "safety" was added, framed as a "health and safety issue" and participants were asked to rank order these issues from most to least important to them.

  1. In lines 291-300, it is better to write the three steps together or write them in sections.

Thank you for this suggestion. The section has been updated as follows: The following four steps were presented: assess, act, communicate, repeat, as depicted in Figure 2, where a continuous improvement risk management approach was discussed.

  1. In line 332, please indent before the paragraph.

Thank you for this suggestion. This has been updated

  1. In the Demographics section, it is better to introduce the participant's allocation criteria in the section of Participant.

Thank you for this suggestion. The following has been removed from the methods section and moved the results section before the demographics table: Of the 544 participants, 273 spent longer than 20 minutes on the survey. Of the 273 participants, N= 267 fully answered the pre-intervention questions and N= 241 fully answered both pre and post-intervention questions, and are the basis of this analysis.

  1. Please uniform the format of Table 1.

Thank you for this suggestion. This has been updated.

  1. Where is the limitation of the study? Please add in the section of Conclusions.

Thank you for this suggestion. Section 4.1 Limitations has been added prior to section 5 Conclusions.

Reviewer 2 Report

I have some questions about methdology:

How 544 participants were selected? How many participants were recruited? How you reached out to recruit them? When (months) did they complete the survey.

If participants were not financially compensated for participation, what is their motivation to watch a 20-minute intervention video and then answer 34 questions?

How many questions in our final questionnaire? ? How those questions were design and developed on the questionnaires?

Author Response

Dear reviewer,

We would like to thank you for your valuable feedback. We believe your input has helped us to improve the manuscript. Below are responses to your feedback:

  1. How 544 participants were selected? How many participants were recruited? How you reached out to recruit them? When (months) did they complete the survey.

Thank you for the opportunity to clarify. Section 2.1 has been updated to include the following information: Participants were recruited via email request from Indiana State University and Indiana University of Pennsylvania mailing lists and from the principal investigator's social media posts via LinkedIn and Facebook.

  1. If participants were not financially compensated for participation, what is their motivation to watch a 20-minute intervention video and then answer 34 questions?

Thank you for the opportunity to clarify. Of the 544 participants who began the survey, only 241 spent adequate time on the survey and fully answered both pre and post-intervention questions. When asked a question on whether participants were interested in learning more about climate change, the majority of respondents indicated they were interested in learning more either through their workplace, community, or both. The authors suggest that those who participated in the study may have been interested in the content and that is why they spent the time. However, participant motivation for participating in the study was not tested.

  1. How many questions in our final questionnaire? ? How those questions were design and developed on the questionnaires?

Thank you for the opportunity to clarify. The survey consisted of thirty-four questions including three demographic questions on participants’ country, state/province, and employment industry and eight worldview questions, see Appendix A. Qualtrics XM was used to collect the data. The data collection section has been updated as follows:

            An original online survey instrument was used consisting of thirty-four questions including three demographic questions on participants’ country, state/province, and employment industry and eight worldview questions, see Appendix A. Participants were asked questions to characterize their worldview based on Cultural Cognition Theory [27] as well as questions to assess their belief that climate change is human-caused (anthropogenic), the extent of individuals’ knowledge of climate change impacts, belief that climate change personally affects them, the importance of the context in which they view climate change (as an environmental issue, a health and safety issue, a scientific issue, an agricultural issue, a political issue, an ethical issue, and an economic issue), whether they believe humans can resolve the climate crises, their motivation to take action, what major context motivates them to act (to protect health and safety of current and future generations, to protect ecosystems and wildlife, and to protect the economy), what are their perceived barriers to action, and what actions they are currently taking and are planning to take to prevent climate change.
            After answering the pre-intervention questions, participants watched a 20-minute intervention video. The pre-intervention questions on anthropogenicity belief, knowledge of climate change impacts, belief that climate change personally affects them, the importance of the context in which they view climate change, belief that humans can resolve the climate crises, their motivation to act, what major context motivates them to act, perceived barriers to action were then repeated. In addition, participants were asked what mitigating actions they were willing to take within the near term (over the next 90 days), and what actions they were willing to take at some time in the future. Participants were also asked if and how the intervention had changed their view and motivation for climate action, whether the information presented had improved their knowledge and awareness of climate change and climate change prevention, as well as their interest in learning more and becoming involved in climate action through their workplace and community.
            A national perception survey "Climate Change in the American Mind" [21] con-ducted by the Yale Program on Climate Communication and George Mason University Center for Climate Change Communication included a survey question on constructs of how Americans “conceptualize” global warming (as an environmental, scientific, severe weather, agricultural, political, economic, health, or ethical/moral/humanitarian issue). The constructs for understanding and prioritizing climate change in the present study were based on this survey question. However, in this study, the construct "safety" was added, framed as a "health and safety issue", and participants were asked to rank order these issues from most to least important to them.
            To assess what major context motivates them to act, participants were provided a selection of three reasons why they might be motivated to mitigate climate change: “to protect the health and safety of current and future generations,” “to protect ecosystems and wildlife, and “to protect the economy.” They were asked to rank order these motivating drivers from most to least important to them.
            Barriers to climate action were based on the literature, particularly Gifford’s [20] “Dragons of Inaction”, where questions were phrased to reflect common pragmatic and psycho-social barriers to taking climate action as follows: “I’m not sure what I can do that will make a difference”, “I can’t afford the carbon-free or carbon-neutral alternative right now”, “Alternatives do not exist or are not practical for my lifestyle”, “I don’t think my family/social group/community group would approve”, “I have a conflict of interest in taking climate change action due to my employment, social or political affiliation”. Participants were asked to rank order these barriers from most to least important to them.
            To assess knowledge of climate change impacts, participants were given thirteen items, based on current and projected impacts from IPCC reports [2] and the United States Fourth National Climate Assessment [4], and asked to select, both pre and post-intervention, all they believe are impacts of climate change. Changes in anthropogenicity belief, belief that climate change affects participants, efficacy belief, and personal motivation to act were measured by asking participants, both pre and post-intervention, to indicate their level of agreement from 0 to 100 where 0 is “not at all” and 100 is “strongly agree” with the following statements: “Climate change is primarily human-caused”, “Climate change affects me”, “Humans can resolve the climate crises”, and the motivation assessment question “I am ready to act now to mitigate (prevent) climate change”. To assess participant worldview, eight out of the 30 survey questions from Kahan’s Cultural Cognition as a Conception of the Cultural Theory of Risk [27] were used, four measuring the Hierarchist/Egalitarian scale and four measuring the Indi-vidualist/Communitarian scale.
            The survey was pilot-tested for face validity among a small sample of participants, N=8, to ensure the survey questions were understood as intended. The pilot-testing confirmed the survey questions and intervention video were understood. Minor changes were made to survey questions to ensure clear language.

Round 2

Reviewer 2 Report

1. Please provide more detailed demographic information of the participants, such as their age, education, race, and income.

2. The limitation section should clarify the conclusion was made by a small sample size (N=241) recruited from the Internet and might not applied to other people with different demographic background.

Author Response

Dear Reviewer,

Thank you for your valuable suggestions. The following are responses to each item.

  1. Please provide more detailed demographic information of the participants, such as their age, education, race, and income.

Thank you for this suggestion. Unfortunately, only information on the participants’ country, state/region, and workplace industry were collected. However, data were collected to categorize participants’ worldview as this has been found in the literature to be the most important factor in shaping personal beliefs and attitudes towards climate change. Under "worldview result, information on worldview category can be found as follows:  

 When grouping participant worldview into categories based on their grid and group scores, many participants provided neutral responses in one or both scales making categorization into 4 quadrants impractical. Based on their responses to the Individualist/Communitarian and Hierarchist/Egalitarian scales, individuals were grouped into the following nine categories: Communitarian/Egalitarian, Communitarian/Hierarchist, Communitarian/Neutral, Individualist/Egalitarian, Individualist/Hierarchist, Individualist/Neutral, Neutral/Egalitarian, Neutral/Hierarchist, and Neutral/Neutral. The majority of participants (110 of the 267 respondents who answered worldview questions) were in the Neutral/Neutral category, 53 participants were in the Communitarian/Neutral cate-gory, 26 were categorized as Individualist/Neutral, 23 were categorized as Communitarian/Egalitarian, 18 were categorized as Neutral/Egalitarian, 12 were categorized as Individualist/Egalitarian, 10 as Individualist/Hierarchist, 10 as Neutral/Hierarchist, and 5 as Communitarian/Neutral, an approximately normal distribution.

  1. The limitation section should clarify the conclusion was made by a small sample size (N=241) recruited from the Internet and might not applied to other people with different demographic background.

Thank you for this suggestion. The limitations section has been updated as follows: While this research contributes to the literature on climate change perceptions, there are limitations to be considered when interpreting the results of this study. Data were cross-sectional so causation cannot be inferred. Data were collected from a relatively small sample size (N=241). There may be selection bias and, while the survey was anonymous, there may be social desirability bias in survey responses. Data were collected using convenience sampling techniques, recruiting participation via social media posts and the use of email lists, which may limit the generalizability of the study results. Political and financial interests of the respondents and their means to overcome action barriers are also limitations that might affect the generalizability of this study. However, when grouping worldview grid and group scales into categories, the distribution of worldview categories was approximately normal, which may support the generalizability of the results. The results likely reflect an American perspective vs a global perspective, particularly a Midwestern perspective, as the majority of participants were from this region in the United States. Although participant responses were excluded from anyone who spent less than 20 minutes on the survey (a minimum time amount needed to fully experience the intervention), evaluating active viewing or engagement was not a capability of the online survey software. Replication of this study in a live setting or use of engagement tracking software in an online setting would control for this limitation.